# Work-Related Identity Discrepancy and Counterproductive Work Behavior: The Role of Emotional Exhaustion and Supervisor Incivility

**DOI:** 10.3390/ijerph17165747

**Published:** 2020-08-09

**Authors:** Chang-E Liu, Xiao Yuan, Chenhong Hu, Tingting Liu, Yahui Chen, Wei He

**Affiliations:** 1Mobile E-business Collaborative Innovation Center of Hunan Province, Key Laboratory of Hunan Province for Mobile Business Intelligence, College of Business Administration, Hunan University of Technology and Business, Changsha 410205, China; liuce15@hutb.edu.cn; 2College of Business Administration, Hunan University of Technology and Business, Changsha 410205, China; 201810310029@stu.hutb.edu.cn (X.Y.); 201910310082@stu.hutb.edu.cn (C.H.); 3School of Business and Tourism Management, Yunnan University, Kunming 650091, China; 4Scott College of Business, Indiana State University, Terre Haute, IN 47809, USA; whe@indstate.edu

**Keywords:** work-related identity discrepancy, counterproductive work behavior, emotional exhaustion, supervisor incivility

## Abstract

This research investigates the role of emotional exhaustion and supervisor incivility in explaining the relationship between work-related identity discrepancy and counterproductive work behavior. Based on resource conservation theory, our study hypothesizes a moderated mediation model that work-related identity discrepancy impacts counterproductive work behavior through emotional exhaustion, and supervisor incivility is deemed as the boundary condition in the indirect effect. Drawing on a sample of 863 employees, we found support for the moderated mediation model in which the positive relationship between work-related identity discrepancy and counterproductive work behavior was mediated by emotional exhaustion, such that the mediating relationship was strengthened for new leaders with a low level of supervisor incivility and weakened for those with high level of supervisor incivility. We further discuss the theoretical and practical implications of these findings.

## 1. Introduction

Leadership change influences a variety of organizational outcomes as new leaders usually differ from their predecessors in leadership styles and strategies [1,2]. For example, the passing of Steve Jobs in 2011 brought about a huge shock to Apple Inc. and led to the departure of several key executives since his successor, Timothy D. Cook, pays more attention to marketing than innovation. Previous research indicated that leadership transition can not only harm the atmosphere and performance of an organization [3,4], but also increase the turnover intention of its employees [5]. In particular, employees who hold a high-quality leader–member exchange relationship with their previous leader may experience more identity loss after leadership transition [6,7] since they were insiders of the previous leader’s “circle” and granted preferential treatment in terms of opportunities and resources in the workplace [8]. Once the old leader leaves, these employees may feel the discrepancy in identity and resources.

Work-related identity discrepancy (or identity loss) refers to the phenomenon that external events break the cognitive balance of existing work-related identity and lead to a gap between one’s actual self and the ideal or ought self [9]. Previous studies have shown that work-related identity discrepancy has negative relationships with employee happiness [10], job satisfaction [11], and job engagement [12], but positive relationships with employee emotional anxiety [13,14] and depression [15,16]. However, these studies mainly focused on the impact of work-related identity discrepancy on employees’ emotions, attitudes, or positive work behaviors but paid little attention to employees’ negative behaviors, such as turnover, absenteeism, insubordination, and so on. Our study attempts to fill this research gap by focusing on the influence of work-related identity discrepancy on a specific negative conduct—employee counterproductive work behavior—since curbing these disruptive behaviors in the workplace is also an important goal for employee performance management.

As a deviant behavior widely existing in organizations [17], counterproductive work behavior refers to an employee’ behavior that intentionally endangers the interests of the organization or its members, including slack, sabotage, theft, and so on [18,19]. Previous research has demonstrated that identity discrepancy or loss affects individuals’ cognitions, emotions, and behaviors [13], and triggers such negative emotions as disappointment, sadness, and fear [20,21], further resulting in negative behaviors [21]. Along this line, we speculate that work-related identity discrepancy is likely to impact counterproductive work behavior. This is another area where our study can fill up the existing research gap and make unique contribution to the literature. 

Moreover, resource conservation theory [22] holds that individuals incline to acquire and preserve resources, such as objects, personal characteristics, conditions, time, and energy; the loss or the threat to loss of resources will cause stress on individuals and lead to various behavioral consequences. Emotions are a vital personal resource that can be consumed or lost. That is, when good emotions are used up, it will lead to emotional exhaustion, a typical stress reaction and a symptom of job burnout [23]. Resource conservation theory also argues that environmental changes often cause or threat the loss of resources [22]. Thus, when a leader leaves, employees who maintain a high-quality leader-member exchange relationship with him or her will not only perceive the discrepancy in work-related identity, but also experience stress and emotional exhaustion due to the potential or real loss of their privileged resources associated with the previous leader [24], subsequently engaging in counterproductive work behavior [25,26]. Therefore, we anticipate that emotional exhaustion plays a mediating role between work-related identity discrepancy and counterproductive work behavior and would like to test this hypothesis with the present empirical study.

As a crucial situational variable, leadership has a significant impact on employee’s emotion [27]. For example, destructive leadership was found to have a stronger positive association with emotional exhaustion than constructive leadership [28]. In the workplace, supervisor incivility is a typical destructive leadership behavior that demonstrates the supervisor’s low-intensity deviant behavior with ambiguous intent to harm the subordinates, in violation of workplace norms for mutual respect [29]. Thus, we propose that employees may feel deep emotional exhaustion when they perceive work-related identity discrepancy and experience uncivil behavior imposed on them by the new leader. That is, supervisor incivility moderates the relationship between work-related identity discrepancy and emotion exhaustion. This is our last research proposition.

In sum, based on resource conservation theory, our research explores the mediating effect of emotional exhaustion on the relationship between work-related identity discrepancy and counterproductive work behavior and the moderating effect of supervisor incivility on this indirect effect. It makes several theoretical contributions to the management literature. First, we not only expand the understanding of the outcomes of work-related identity discrepancy, but also reveal its diverse effects. Second, we probe emotional exhaustion as a psychological mechanism underlying the impact of work-related identity discrepancy on employee counterproductive work behavior. Last, we introduce the new boundary condition of supervisor incivility into the study of work-related identity discrepancy and counterproductive work behavior.

## 2. Theoretical Background

### 2.1. Work-Related Identity Discrepancy and Counterproductive Work Behavior

Having experienced leadership change, employees who maintain a high-quality leader-member exchange relationship with the previous leader will perceive work-related identity discrepancy [7,11]. Previous research suggests that individuals tend to generate negative emotions (e.g., disappointment, sadness, and fear) when they find a discrepancy between their actual self and their ideal self or ought self [20,30]. Negative work events involving interactions with leaders have both direct and indirect positive effects on negative emotions [21]. Furthermore, employees’ negative emotions will positively affect workplace deviant behaviors such as counterproductive work behavior [21,31,32]. Therefore, when employees experience leadership change as a negative work event, they will perceive a rise in work-related identity discrepancy, which may have a positive effect on employees’ counterproductive work behavior.

Moreover, resource conservation theory pointed out that people will try their best to maintain and protect their existing resources, avoid loss of resources, and create and acquire new resources [22]. When leadership changes, employees who maintain high-quality leader-member exchange relationship with the previous leader could lose their "insider" identity and advantageous resources of materials and emotions granted by their previous leader [23]. Hence leadership change will let employees feel the work-related identity discrepancy, which in turn leads to the dual loss of their material resources and emotional resources. Moreover, in order to “get even” or minimize the loss of resources, employees might display counterproductive work behaviors, such as slack, sabotage, or even theft [19]. Based on the discussion, we propose our first hypothesis:

**Hypothesis** **1.**
*Work-related identity discrepancy has a positive effect on counterproductive work behavior.*


### 2.2. Mediating Effect of Emotional Exhaustion

Emotional exhaustion refers to a psychological state of physical and emotional exhaustion caused by stress, a crucial dimension of job burnout [33]. Changes in employees’ perceived work-related identity can lead to changes in their emotions [34]. In other words, when facing leadership change, employees who maintain a high-quality leader-member exchange relationship with the previous leader will perceive work-related identity discrepancy and experience negative emotions [20]. For example, the work-related identity discrepancy between "actual vs. ideal" self is related to depression, whereas the "actual vs. ought" discrepancy is related to agitation [20]. Increased negative emotions will reduce employees’ positive emotions and eventually lead to emotional exhaustion [35]. 

Previous studies have also shown that employees’ emotional state is closely related to their behavior [36,37,38]—emotional exhaustion has direct positive effects on employee counterproductive work behavior [25]; it is actually an essential antecedent of counterproductive work behavior [26]. More recent research also showed that employees are more likely to carry out counterproductive work behavior when they experience emotional exhaustion [39]. 

On the other hand, resource conservation theory has demonstrated that sufficient resources can reduce individuals’ stress, but the loss of resources will increase their stress and trigger negative behaviors [40,41]. When facing leadership change, employees who maintain a high-quality leader-member exchange relationship with the previous leader may perceive work-related identity discrepancy and loss of material or emotional resources, which ultimately cause the employees’ stress and tension. Consequently, the stress and tension will exacerbate the consumption of employees’ emotional resources and further lead to emotional exhaustion [42]. Therefore, employees might engage in counterproductive work behavior to relieve their stress and reduce resource loss. Based on these inferences, we propose our second hypothesis:

**Hypothesis** **2.**
*Emotional exhaustion mediates the effect of work-related identity discrepancy on counterproductive work behavior.*


### 2.3. Moderating Effect of Supervisor Incivility

Existing research indicates that situation plays a significant role in moderating the effects of individual differences on individual behaviors [43]. Leadership, both constructive and destructive, is such a critical situational factor in the organization [27]. Meanwhile, people are found to be more responsive to the negative aspects of their external context than to the positive ones across a broad range of psychological phenomena; that is, negative contextual aspects tend to have stronger influences on their attitude and behavior than the positive ones [35]. That means, as a typical destructive leadership, supervisor incivility may play a significant role in moderating the effects of individual differences (such as work-related identity discrepancy) on individual behaviors (such as counterproductive work behavior).

We theorize that supervisor incivility, as a destructive leadership, will reduce the positive relationship between work-related identity discrepancy and employee emotional exhaustion. On the one hand, leadership change makes those employees who maintain a high-quality leader-member exchange relationship with the previous leader feel their identity discrepancy and further experience negative emotions [20]. The increased negative emotions will reduce employees’ positive emotions and eventually lead to emotional exhaustion [44]. On the other hand, if the new leader displays incivility, it will constitute a strong situation that reduces the influence of work-related identity discrepancy on emotional exhaustion [43] since supervisor incivility, such as public criticism, slanders, or satirizing subordination, is a crucial job stressor beyond employees’ control that can exacerbate their emotional exhaustion [29,45]. Thus, we suggest that the influence of work-related identity discrepancy on employee emotional exhaustion is narrowed in strong situations (namely, the new leader engages in intensive incivility).

According to resource conservation theory [22], individuals are strongly committed to acquiring and securing resources, but their physical, emotional, and cognitive resources are usually limited. Thus, when the new leader engages in high-intensity incivility, it may take up the employees’ psychological resource consumption and aggravate their emotional exhaustion to cope with it. In this situation, the employees’ emotional exhaustion is mainly affected by the new leader’s incivility. In contrast, when the new leader displays low-intensity incivility, employees’ emotional exhaustion is mostly initiated by work-related identity discrepancy. Thus, we propose the following moderating hypothesis:

**Hypothesis** **3.**
*Supervisor incivility moderates the positive effect of work-related identity discrepancy on emotional exhaustion, such that the positive effect of work-related identity discrepancy on emotional exhaustion is stronger when supervisor incivility is low but weaker when supervisor incivility is high.*


### 2.4. Moderated-Mediation Model

Hypothesis 1 proposes a direct effect of work-related identity discrepancy on counterproductive work behavior; Hypothesis 2 suggests a mediating effect of emotional exhaustion; and Hypothesis 3 offers a moderating effect of supervisor incivility. Following the logic of these three previous hypotheses and drawing on Hayes’ [46] recommendation on the moderated mediation effect, we set forth a moderated-mediation model as follows:

**Hypothesis** **4.**
*The indirect positive effect of work-related identity discrepancy on counterproductive work behavior through emotional exhaustion is moderated by supervisor incivility.*


Altogether, we summarize our research variables and hypotheses in a conceptual framework in Figure 1.

## 3. Method

### 3.1. Participants and Procedures

Our research proposal had been approved by the Academic Ethics Committees of our institutions before we began collecting data (IRB 870-636). Participants of our research were full-time employees and managers randomly sampled from 24 organizations in manufacturing, high tech, services, government, and non-profit sectors in southern China. All the participants had experienced leadership change (or change of boss) in their workplace before. We asked the human resources personnel in those organizations to help us distribute and collect our survey questionnaires onsite and online during business hours. Our questionnaire explained to all participants of voluntary participation and anonymity of the survey. In total, 1074 copies of questionnaire were distributed and 863 valid copies were collected (response rate = 80.4%). Of the 863 valid respondents, 43.8% were men and 56.2% women. Their average age was 30.14 years (SD = 6.51, range 18–59). With regard to education level, 24.1% had a master’s degree or above, 52.96% had a bachelor’s degree, and 22.94% had a high school diploma or below. As to their position, 35.11% were rank-and-file employees, 26.54% supervisors, 23.75% middle-level managers, and 14.60% senior managers. In terms of tenure, 45.54% had less than 2 years of work experience, 29.90% with 2–5 years, 15.87% with 5–10 years, and 8.69% with 10 or more years of work experience.

### 3.2. Measures 

To ensure the reliability and validity of measurements, we adopted well-established scales developed and used by previous researchers. All the scales were initially developed in English and then translated into Chinese through back-translation validation [47]. They all used Likert 5-point scale rated from 1 (strongly disagree) to 5 (strongly agree) except those demographic questions (i.e., gender, age, education, position, and tenure).

Work-related identity discrepancy: We measured work-related identity discrepancy with a 3-item scale developed by Khan, Moss, Quratulain, and Hameed [48]. A sample item is “After experiencing replacement of leadership, I felt that my identity was threatened” (α = 0.900).

Emotional exhaustion: We adopted a 3-item scale developed by Watkins, Ren, Umphress, Boswell, Triana, and Zardkoohi [49] to measure emotional exhaustion. A sample item is “I feel emotionally drained from my work.” (α = 0.903).

Counterproductive work behavior: We used a 17-item scale developed by Zhang [50] to measure counterproductive work behavior. A sample item is “After the leadership change, some of my colleagues do not work very hard.” (α = 0.976).

Supervisor incivility: We measured supervisor incivility with a 7-item workplace incivility scale developed by Cortina, Magley, Williams, and Langhout [51]. A sample item is “The new manager or supervisor put me down or was condescending to me” (α = 0.954). 

Control variables: To ensure the accuracy and rigor of our results and rule out alternative explanations, we used the respondents’ gender, education, age, position, and tenure as the control variables to observe their impacts on counterproductive work behavior [52].

### 3.3. Data Analyses 

We used SPSS 22.0 (IBM Corp., Armonk, NY, USA), PROCESS macro (The Ohio State University, Columbus, OH, USA), and MPLUS 7.0 (Muthen & Muthen, Los Angeles, CA, USA) to analyze the data and test the hypotheses. We first analyzed the descriptive statistics of all variables; then, we examined the common method bias; and lastly, we tested all the research hypotheses: the main effect, the mediating effect, the moderating effect, and the moderated mediation model.

## 4. Results

### 4.1. Preliminary Analyses

The correlation coefficients, means, and standard deviations of all variables are shown in Table 1. As seen in the table, work-related identity discrepancy has a significantly positive correlation with emotional exhaustion (γ = 0.575, *p* < 0.01) and counterproductive work behavior (γ = 0.531, *p* < 0.01), respectively; emotional exhaustion and counterproductive work behavior have a significantly positive correlation (γ = 0.547, *p* < 0.01). These results preliminarily support the hypotheses of the main effect and mediating effect. Moreover, all coefficients were smaller than 0.70, indicating that there was no multi-collinearity problem with our data.

Following Zhou and Long’s [53] suggestions, we first conducted a varimax rotation analysis of principal factors for all four latent variables—work-related identity discrepancy, emotional exhaustion, supervisor incivility, and counterproductive work behavior—so as to examine the presence and magnitude of common method variance, according to the number of factor precipitation or common factor interpretation. Four common factors (eigenvalue > 1) were extracted from the test results, and the first factor explained only 38.94% of the variance, that is, less than the recommended explanation criterion of 50%. Therefore, we concluded that common method variance in the present research was not significant.

Then, we conducted a series of confirmatory factor analyses using MPLUS 7.0 [54] to ensure that the four latent variables all have satisfactory discriminant validity. The confirmatory factor analysis results in Table 2 demonstrate that our hypothesized four-factor model is a better fit to the data (x^2^/df = 3.172 < 4, RMSEA = 0.050 < 0.08, IFI = 0.967, TLI = 0.964, CFI = 0.967 > 0.9) than those more parsimonious models: a three-factor model (M1) with work-related identity discrepancy and emotional exhaustion loaded on one factor (x^2^/df = 5.476, RMSEA = 0.072, IFI = 0.931, TLI = 0.925, CFI = 0.931); a three-factor model (M2) with supervisor incivility and emotional exhaustion loaded on one factor (x^2^/df = 5.720, RMSEA = 0.074, IFI = 0.927, TLI = 0.921, CFI = 0.927); a two-factor model (M3) with work-related identity discrepancy, counterproductive work behavior, emotional exhaustion, and supervisor incivility loaded on one factor x^2^/df = 8.918, RMSEA = 0.096, IFI = 0.878, TLI = 0.868, CFI = 0.877); and a one-factor model (M4) with all variables loaded on a single factor (x^2^/df = 18.614, RMSEA = 0.141, IFI = 0.734, TLI = 0.714, CFI = 0.734).

### 4.2. Main Effect, Mediating Effect, and Moderating Effect Testing 

Our study used SPSS 22.0 to verify the main effect. As shown in Table 3, we first entered the control variables (gender, age, education, position, and tenure) into the regression model and then work-related identity discrepancy via stepwise. The results show a significantly positive correlation between work-related identity discrepancy and counterproductive work behavior (M1, β = 0.481, *p* < 0.001). Thus, Hypothesis 1 is supported.

Following the procedure suggested by Preacher and Hayes [55], we tested the mediating effect with PROCESS macro with 5000 bootstrap samples and a confidence interval (CI) of 95%. The indirect effect of work-related identity discrepancy on counterproductive work behavior via emotional exhaustion is 0.172, with a 95% confidence interval (CI) [0.129, 0.218], not including 0 (not shown in Table 3). That is, the indirect effect is significant. Therefore, Hypothesis 2 is supported.

Next, we tested the moderating role of supervisor incivility. To reduce potential collinearity between work-related identity discrepancy and supervisor incivility, we decentralized all explanatory variables (except demographic variables) [56]. As shown in Table 3, work-related identity discrepancy has a significantly positive correlation with emotional exhaustion (M3, *β* = 0.535, *p* < 0.001), after entering the interaction term between work-related identity discrepancy and supervisor incivility in Model 5, and the interaction coefficient is significant (β = −0.173, *p* < 0.001), R^2^ = 0.405 (*p* < 0.001). These results indicate that supervisor incivility played a negative moderating role in the relationship between work-related identity discrepancy and emotional exhaustion. Therefore, Hypothesis 3 receives support. That is, when supervisor incivility is high, the positive correlation between work-related identity discrepancy and emotional exhaustion is weak. In addition, after all variables being decentralized, the effect of supervisor incivility on emotional exhaustion (M4, β = 0.596, *p* < 0.001) is still stronger than that of work-related identity discrepancy on emotional exhaustion (M3, β = 0.535, *p* < 0.001), which is consistent with our inference.

To make the moderating effect of supervisor incivility look more intuitive and specific, we followed Preacher, Curran, and Bauer’s [57] approach and drew two schematic diagrams. As shown in Figure 2, when supervisor incivility is high, the positive relationship between work-related identity discrepancy and emotional exhaustion is weak (simple slope = −0.183, *p* < 0.001). On the contrary, when supervisor incivility is low, the positive relationship between work-related identity discrepancy and emotional exhaustion is strong (Simple slope = 0.747, *p* < 0.001).

### 4.3. Moderated-Mediation Effect Testing

We further bootstrapped the confidence interval to assess supervisor incivility’s conditional effect on the relationship between work-related identity discrepancy and counterproductive work behavior via emotional exhaustion. As shown in Table 4, when supervisor incivility is low, the indirect effect is significant (indirect effect = 0.067, 95% confidence interval = 0.025 to 0.112); when supervisor incivility is high, the indirect effect is still significant (indirect effect = 0.012, 95% confidence interval = 0.001 to 0.031). These results suggest that supervisor incivility moderates the indirect effect of work-related identity discrepancy on counterproductive work behavior through emotional exhaustion. Nevertheless, according to Hayes’ suggestion [46], when the indirect effects are significant, no matter how high or low the moderator is, the index criterion must be applied to determine whether the moderated-mediation effect is significant. In Table 4, the index of supervisor incivility is −0.019, CI (−0.033, −0.007), not including 0. Thus, Hypothesis 4 is supported.

## 5. Discussion

Our study focuses on the effect of work-related identity discrepancy on counterproductive work behavior, especially the mediating role of emotional exhaustion and the moderating role of supervisor incivility. We found that: (1) work-related identity discrepancy has a significant positive effect on counterproductive work behavior; (2) emotional exhaustion significantly mediates the main effect between work-related identity discrepancy and counterproductive work behavior; (3) supervisor incivility plays a moderating role between work-related identity discrepancy and emotional exhaustion, specifically, the relationship between work-related identity discrepancy and emotional exhaustion decreases as supervisor incivility increases; and (4) supervisor incivility further influences the positive indirect relationship between work-related identity discrepancy and counterproductive work behavior via emotional exhaustion: the mediating role of emotional exhaustion weakens when supervisor incivility is high and strengthens when it is low. We further discuss the theoretical and practical implications of these findings.

### 5.1. Theoretical Implications

Our research contributes to the existing management literature in three aspects. First, previous studies on work-related identity discrepancy focused more on its effects on emotions, attitudes, and perceptions [11,13,14] and less on employee behaviors, which are more directly related to organization performance. Our study not only enriches the consequences of work-related identity discrepancy but also sheds a light on the dark side of work-related identity discrepancy by examining its effect on counterproductive work behavior. In addition, our research provides a new perspective for identity research from the field of management study, responding to the call that future research should pay more attention to the employees’ negative attitudes and behaviors resulting from work-related identity maladjusted [30].

Second, our research also reveals the specific psychological mechanism how work-related identity discrepancy is related to counterproductive work behavior. Drawing on resource conservation theory, our research uncovers that employees’ emotional resources can be consumed by work-related identity discrepancy and lead to counterproductive work behavior as a reaction to workplace stress and job burnout. Meanwhile, existing studies on emotional exhaustion mainly focused on its antecedents (such as role ambiguity, role conflict, work-family conflict, supervisory support, work overload, and perceived organizational support) and consequences (such as job satisfaction, organizational commitment, turnover intentions, and job performance) [58] while largely ignoring the role of mediation. Thus, our finding on the mediation effect of emotional exhaustion not only broadens the advocacy that emotional factors can explain the process of how people deal with work-related identity discrepancy after leadership change, but it also suggests that identity loss can be another source of emotional exhaustion like stress, personal characteristics [59], interpersonal milieu [60], and work itself [58]. 

Last, we demonstrate the important role of supervisor incivility between work-related identity discrepancy and counterproductive work behavior. We found that the mediating effect of the emotional exhaustion on counterproductive work behavior through work-related identity discrepancy is moderated by supervisor incivility, and supervisor incivility can substitute for the positively mediated effect of work-related identity discrepancy on counterproductive work behavior since it reduces individual variation in emotional exhaustion and hence its mediating role. This finding supports the viewpoint the effects of individual variables are reduced in a strong situation [61] because supervisor incivility represents such a strong situation that employees try to avoid. Moreover, our finding is in line with previous research by Holm, Torkelson, and Bäckström [62], who found that merely being in a climate of supervisor incivility can affect employees’ deviant behavior. 

### 5.2. Practical Implications

Our study also provides important implications for organization management in practice. Our findings show that employees experience more emotional exhaustion because of work-related identity discrepancy and thus engage in more counterproductive work behavior. Meanwhile, employees who had a close relationship with the previous leader experienced more work-related identity discrepancy. Therefore, when a leader departs, the organization should pay more attention to the "insiders" of the leader and try to comfort them, alleviate their emotional exhaustion, and reduce the negative impact of the leader’s departure on their performance. Moreover, the organization should set leadership accountability or departure due diligence policies to reduce the impact of destructive leadership on employees. Our findings also indicate that supervisor incivility has a significant substitution effect for work-related identity discrepancy on counterproductive work behavior, implying that the new leader plays a more important role than work-related identity discrepancy during the leadership transition. Therefore, organizations should cautiously select their new leaders and take into account their character as well as their past performance and future potential. 

Last but not least, our research shows that emotional exhaustion plays a mediating role between work-related identity discrepancy and counterproductive work behavior. Therefore, when facing the loss of emotional resources caused by leadership transition, employees should be encouraged to participate in various employee wellbeing or employee assistance programs sponsored by the employer, such as stress management seminar, coaching and counselling, social support group, or other recreational or entertaining activities. These constructive practices can help improve employees’ mental health, develop or preserve their emotional resources, and mitigate their stress and burnout brought in by the departure of their previous leader.

### 5.3. Limitations and Directions for Future Research

Like any other empirical research in this area, our study also has several limitations. First, we only collected the cross-sectional data and ignored the time effect on variables. This prevented us from testing the dynamic impact of work-related identity discrepancy on counterproductive work behavior, although our conclusion demonstrated that work-related identity discrepancy could be deemed as a predictor of counterproductive work behavior. Nevertheless, there is still a need to revalidate our findings with data from multiple sources and multiple times to get more robust results. 

Second, although our research explored the effect of boundary condition of supervisor incivility and found an interesting fact that supervisor incivility has a significant substitution effect for work-related identity discrepancy on counterproductive work behavior, other destructive leadership behaviors, such as abusive supervision, may also exhibit a similar pattern of moderating effect. Thus, researchers in the future can take other destructive leadership behaviors including abusive supervision into consideration and study how they influence the effect of work-related identity discrepancy on employee behaviors. 

Last, we did not test other possible theories for our results. Even though we felt that resource conservation theory is the most suitable to explain the relationship between work-related identity discrepancy and counterproductive work behavior, alternative theories could also be adopted to explain the phenomenon, such as affective event theory [63], which suggests that employees’ emotional responses stemming from specific work events will eventually affect their attitudes and behaviors. Therefore, future research can expand the research area by designing other schemes using several alternative theories in explaining the effect of work-related identity discrepancy on employee behaviors through employee emotions and finally testing the relative strengths of several alternative theories.

## 6. Conclusions

Our study draws on resource conservation theory to examine the role of emotional exhaustion and supervisor incivility in explaining the relationship between work-related identity discrepancy and counterproductive work behavior. This not only enriches understanding of the outcomes of work-related identity discrepancy, such as counterproductive work behavior, but also expands the research on leadership change and employee emotion. We hope our study can serve as a springboard for future research in relevant areas.

## Figures and Tables

**Figure 1 ijerph-17-05747-f001:**
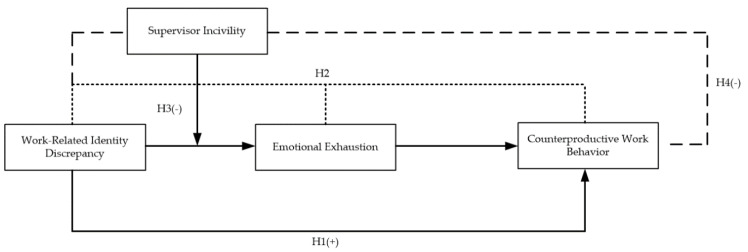
The research conceptual model.

**Figure 2 ijerph-17-05747-f002:**
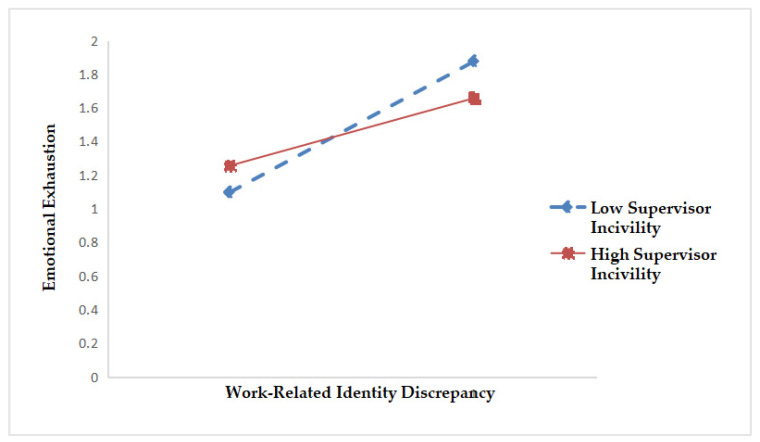
The moderating effect of supervisor incivility on the relationship between work-related identity discrepancy and emotional exhaustion.

**Table 1 ijerph-17-05747-t001:** The descriptive statistical analysis and correlations (N = 863).

Variable	M	SD	1	2	3	4	5	6	7	8	9
1. Gender	1.56	0.49									
2. Age	30.14	6.51	−0.140 **								
3. Education	2.97	0.98	0.022	0.187 **							
4. Position	2.82	1.07	0.030	−0.115 **	0.197 **						
5. Tenure	2.70	1.19	−0.091 **	0.444 **	0.115 **	−0.104 **					
6. WRID	3.16	1.23	0.002	0.143 **	−0.064	−0.178 **	−0.177 **	(0.900)			
7. EE	3.01	1.25	0.056	0.085 *	−0.079 *	−0.155 **	−0.176 **	0.575 **	(0.903)		
8. SI	3.05	1.20	0.012	0.165 **	−0.054	−0.191 **	−0.162 **	0.579 **	0.626 **	(0.954)	
9. CWB	3.14	1.13	−0.008	0.145 **	−0.052	−0.158 **	−0.139 **	0.531 **	0.547 **	0.625 **	(0.976)

WRID = Work-Related Identity Discrepancy; EE = Emotional Exhaustion; SIm= Supervisor Incivility; CWB = Counterproductive Work Behavior. Reliabilities (Cronbach’s α) are on the diagonal in parentheses. As for gender, men are coded as “1” and women as “2”. ** *p* < 0.01, * *p* < 0.05.

**Table 2 ijerph-17-05747-t002:** Comparison of measurement models.

Model	*x*^2^/df	RMSEA	IFI	TLI	CFI
Baseline model	3.172	0.050	0.967	0.964	0.967
M1	5.476	0.072	0.931	0.925	0.931
M2	5.720	0.074	0.927	0.921	0.927
M3	8.918	0.096	0.878	0.868	0.877
M4	18.164	0.141	0.734	0.714	0.734

**Table 3 ijerph-17-05747-t003:** The main and moderating effects (N = 863).

Variables	CWB		EE			
	M1	M2		M3	M4	M5
Control						
Gender	0.009	0.001		0.056 *	0.047	0.045
Age	0.253 ***	0.122 ***		0.063	0.037	0.002
Education	−0.041	−0.019		−0.038	−0.039	−0.036
Position	−0.148 ***	−0.067 *		−0.060 *	−0.042	−0.016
Tenure	−0.260 ***	−0.112 **		−0.105 **	−0.090 **	−0.044
Independent						
WRID		0.481 ***		0.535 ***		0.301 ***
Moderator						
SI					0.596 ***	0.418 ***
Interaction						
WRID×SI						−0.173 ***
F	18.686	61.197		75.871	96.938	105.170
R^2^	0.098 ***	0.300 ***		0.347 ***	0.405 ***	0.497 ***
△R^2^	0.093	0.296		0.343	0.401	0.492

WRID = Work-Related Identity Discrepancy; EE = Emotional Exhaustion; SI = Supervisor Incivility; CWB = Counterproductive Work Behavior. As for gender, men are coded as “1” and women as “2”. *** *p* < 0.001, ** *p* < 0.01, * *p* < 0.05.

**Table 4 ijerph-17-05747-t004:** The moderated-mediation effect.

Moderator SI	WRID(X)→EE(M1)→CWB(Y)							
	Conditional Indirect Effect				Moderated-Mediation Effect			
	Effect	SE	ULCI	LLCI	Index	SE	ULCI	ULCI
Low SI	0.067	0.022	0.025	0.112	−0.019	0.007	−0.033	−0.007
High SI	0.012	0.008	0.001	0.031				

WRID = Work-Related Identity Discrepancy; EE = Emotional Exhaustion; SI = Supervisor Incivility; CWB = Counterproductive Work Behavior.

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
