# Peer review of "Work-Related Identity Discrepancy and Counterproductive Work Behavior: The Role of Emotional Exhaustion and Supervisor Incivility"

_ijerph, 2020, doi:10.3390/ijerph17165747_

Round 1
Reviewer 1 Report
The authors have explored the role of emotional exhaustion and supervisor incivility in the relationship between work-related identity discrepancy and counterproductive work behavior.
The article is well written and it shows proper rationale. All statistical analyses are properly conducted. However, I have a main concern:
One of the main tested variables is emotional exhaustion, which is a symptom of job burnout. The authors pay little to none attention to relevant and important personal resources associated both emotional exhaustion, job burnout and work behavior, namely emotional intelligence, optimism, core-self evaluations, resilience, gratitude, forgiveness and health-promoting behaviors. Although these personal resources are not included in the study design, I consider that they should be at least mentioned in the introduction and also in the discussion (in "theoretical implications" and especially in "practical implications"). The authors state that workout and tourism would "supplement (..) good personal resources". Any reader minimally familiarized with personal resources would find this statement very weak. Authors could refer to preventive programs aimed to increase employees´emotional intelligence, and to boost other personal resources like optimism, resilience, gratitude and forgiveness, as well as health-promoting behaviors (like health responsibility, nutrition, physical activity, interpersonal relationships/social support, stress management and spiritual growth) as potential strategies to improve emotional resources.
Author Response
Please see the document attached. Thanks.

Reviewer 2 Report
Dear Authors
I carefully evaluated the study, finding it overall interesting, well written and well presented. The theoretical background is clear, but study motivation and literature gap can be stated in a more explicit way.
Participants: I suggest to report the working sectors of the enrolled companies. What was the main jobs of recruited employees? I suggest to describe more in detail the study population, especially about the working and organizational scenario.
Some minor revisions:
Table 1. gender is a dichotomous variable, so it is better to avoid reporting mean and sd for categorical variables.
Figure 1. It is not clear the link for the line starting from H4. I suppose the line should arrive in “supervisor incivility”. Is it correct?
Results are clearly presented.
Discussion section needs some improvements: the comparison between your results and the existing literature is lacking. Briefly, discuss the results of the highlighted relationship in the light of the existing literature about the theme of emotional exhaustion and supervisor incivility.
Author Response

(The authors gave the same response as above.)

Round 2
Reviewer 1 Report
Thank you for the opportunity to again review this manuscript. I appreciate how the authors addressed the concerns related to emotional variables. I consider that this is an important manuscript for the field and adding these emotional considerations strengthens the manuscript and its implications for the field.
Reviewer 2 Report
Dear Authors
The paper has been substantially improved.
Well done.
Best regards